# Supplementation with *Vitis vinifera* Jingzaojing Leaf and Shoot Extract Improves Exercise Endurance in Mice

**DOI:** 10.3390/nu14194033

**Published:** 2022-09-28

**Authors:** Yong Gyun Lee, Hayoung Woo, Chul Choi, Ga-Hee Ryoo, Yun-Jo Chung, Ju-Hyung Lee, Su-Jin Jung, Soo-Wan Chae, Eun Ju Bae, Byung-Hyun Park

**Affiliations:** 1School of Pharmacy, Jeonbuk National University, Jeonju 54896, Jeonbuk, Korea; 2Department of Nutritional Sciences, University of Connecticut, Storrs, CT 06269, USA; 3Department of Neurosurgery, Jeonbuk National University Medical School, Jeonju 54896, Jeonbuk, Korea; 4Advanced Radiation Technology Institute, Korea Atomic Energy Research Institute, Jeongeup 56212, Jeonbuk, Korea; 5Biomedical Research Institute, Jeonbuk National University, Jeonju 54907, Jeonbuk, Korea; 6Department of Preventive Medicine, Jeonbuk National University Medical School, Jeonju 54896, Jeonbuk, Korea; 7Clinical Trial Center for Functional Foods, Jeonbuk National University Hospital, Jeonju 54907, Jeonbuk, Korea; 8Department of Biochemistry and Research Institute for Endocrine Sciences, Jeonbuk National University Medical School, Jeonju 54896, Jeonbuk, Korea

**Keywords:** Jingzaojing, exercise endurance, myofiber, mitochondria, Sirt6

## Abstract

Switching myofibers from the fast-glycolytic type to the slow-oxidative type is associated with an alleviation of the symptoms associated with various cardiometabolic diseases. This study investigates the effect of *Vitis vinifera* Jingzaojing leaf and shoot extract (JLSE), which is rich in phenolic compounds, on the regulation of skeletal muscle fiber-type switching, as well as the associated underlying mechanism. Male C57BL/6N mice were supplemented orally with vehicle or JLSE (300 mg/kg) and subjected to treadmill exercise training. After four weeks, mice in the JLSE-supplemented group showed significantly improved exercise endurance and mitochondrial oxidative capacity. JLSE supplementation increased the expression of sirtuin 6 and decreased Sox6 expression, thereby elevating the number of mitochondria and encouraging fast-to-slow myofiber switching. The results of our experiments suggest that JLSE supplementation reprograms myofiber composition to favor the slow oxidative type, ultimately enhancing exercise endurance.

## 1. Introduction

Skeletal muscle is composed of two distinct types of myofibers that express different forms of myosin heavy chains (MyHC). Slow-twitch fibers that express varying degrees of MyHC-I and MyHC-IIa are rich in mitochondria, depend on oxidative metabolism for their energy, and are resistant to fatigue. Fast-twitch fibers that express MyHC-IIx and MyHC-IIb depend on glycolytic metabolism [1]. Skeletal muscles have the ability to change their fiber types (a phenomenon referred to as muscle plasticity) to adapt to different conditions. For example, endurance exercise increases the number of slow muscle fibers, whereas inactivity triggers fiber type switching from slow-to-fast [2,3]. Accordingly, an increase in oxidative capacity has been shown to decrease the risk factors for cardiometabolic diseases in humans [4]. The clinical importance of fast-to-slow myofiber switching has been recognized for decades. Earlier studies have shown that the proportion of slow oxidative fibers is reduced while the proportion of fast glycolytic fibers is increased in skeletal muscles from patients with obesity and type 2 diabetes [5,6]. The reduction in slow myofibers could account for the decreased oxidative capacity of skeletal muscles and thus increased adiposity in obesity-associated pathogenesis [7,8]. Additionally, the percentage of slow myofibers is positively correlated with insulin-stimulated glucose uptake in human skeletal muscle [9].

Over the last two decades, a number of gain- or loss-of-function studies have identified the signaling molecules that control muscle plasticity. Peroxisome proliferator-activated receptor-*γ* coactivator-1α (PGC-1α) has been isolated as the primary transcriptional regulator that accelerates mitochondrial biogenesis [10]. Treadmill running has been shown to increase the expression of PGC-1α and the development of slow muscle fibers in skeletal muscles [11]. However, as it is difficult for most people to perform regular exercise over a long period, alternative strategies that change muscle fiber types must be found. Several research groups have reported a noticeable improvement in the exercise performance of mice fed diets supplemented with green tea extract [12], resveratrol [13], or probiotics [14,15], though the molecular mechanisms that control the fiber type switching remains to be elucidated. This potentially forms the basis for the use of natural flavonoids to target slow oxidative myofibers in the treatment of metabolic and musculoskeletal diseases [16].

Grapevine is a versatile plant used in a wide range of foods. It can be consumed fresh or in vinification. Grape contains a high amount of polyphenols, including resveratrol and quercetin, and is considered helpful for cardiometabolic diseases [17]. Recent studies have reported the enrichment of phenolic compounds in grape shoots and leaves and their impact on human health [18]. Grape stalk exhibits an anti-oxidant capacity and attenuates obesity, hepatic steatosis, and cardiovascular diseases [19,20]. Grape leaf is rich in anthocyaninins and flavonoids [21] and has been used to treat ethanol-induced liver injury [22], hypertension [23], diabetes [24], and obesity [25]. Additionally, Mihegishi et al. [26] reported that red grape leaf extract increases endurance exercise after swimming by inducing fatty acid oxidation enzymes in skeletal muscles. However, this study did not investigate the phenotypic changes to muscle fibers and molecular pathways that lead to fiber type switching. In this study, we investigated the effect of the ethanol extract of *Vitis vinifera* Jingzaojing leaves and shoots (JLSE) on treadmill exercise-induced endurance and phenotypic changes in the muscle fibers in mice.

## 2. Materials and Methods

### 2.1. Preparation of JLSE

Dried leaves and stems of *V. vinifera* Jingzaojing (100 g) were treated with 1 L of 3% acetic acid-30% ethanol for 3 h at 90 °C. The extracted solution was centrifuged (13,000 rpm, 10 min), and the supernatant was filtered, lyophilized, and powdered.

### 2.2. High-Performance Liquid Chromatography (HPLC) Analysis

The JLSE composition was analyzed using high-performance liquid chromatography (HPLC). An Agilent 1200 Series HPLC (Agilent Technologies, Wilmington, DE, USA) was employed for the analysis. Quercetin-3-O-glucoronide was purchased from Sigma-Aldrich (St. Louis, MO, USA) and used as our reference standard. Aliquots of 10 µL of the processed samples were injected into the HPLC system, which was fitted with a Phenomenex Kinetex C_18_, 50 × 2.1 mm column maintained at 35 °C. A mobile phase composed of 0.1% formic acid in distilled water (Buffer A) and 0.1% formic acid in acetonitrile (Buffer B) was used to separate the analysts with a liner gradient over 20 min at a 0.5 mL/min flow rate. The peaks were detected with UV-VIS detector with 350 nm. Quercetin-3-O-glucoronide content (4.8 mg/g) in the JLSE was calculated based on the relevant peak area using an external standard method.

### 2.3. Animals and Ethical Statements

Male C57BL/6N mice at 20 weeks of age were purchased from Orient Bio (Seoul, Korea) and housed in cages under standard conditions (22 ± 2 °C, 50–60% humidity, 12 h light–dark cycles) throughout the experiment. Mice were fed a standard laboratory chow diet ad libitum. JLSE dissolved in PBS was administered once a day to mice via oral gavage at dose of 300 mg/kg for 28 days. We chose the dose of JLSE based on previous studies in which the oral treatment of grape leaf extract at 200 and 500 mg/kg had beneficial effects against high-fat-diet-induced obesity in mice [19,20] and alcohol-induced liver injury in rats [22]. All animal experiments were performed in accordance with the Guide for the Care and Use of Laboratory Animals published by the US National Institutes of Health (NIH Publication No. 85-23, revised 2011). The study protocol was approved by the Institutional Animal Care and Use Committee of Jeonbuk National University (permit number: JBNU 2021-0125).

### 2.4. Treadmill Running and Grip Strength Measurement

For the treadmill running tests, a single lane treadmill (Jeung Do Bio & Plant, Seoul, Korea) was used. The mice were acclimatized to the treadmill with a daily 30 min run at 10 m/min over 7 days. The mice were then subjected to a daily chronic treadmill running test over a 4 week period. The treadmill initially ran at 10 m/min, but this was increased by 2 m/min every 10 min until 16 m/min was reached, at which point the mice ran until exhaustion (Appendix A). Running time and distance were recorded for each mouse. The forelimb grip strength of mice was measured using a digital grip-strength meter (Jeung Do Bio & Plant, Seoul, Korea) and normalized by lean body mass. Each mouse grabbed a grid and was horizontally pulled by its tail away from the grid as peak force was measured. Each mouse was tested three times, with each test conducted after a 15 min rest period.

### 2.5. Statistical Analysis

The data are expressed as the mean ± standard deviation of the mean (SD). A Kaplan–Meier survival curve was used to estimate the median exhaustion time of mice, and a log-rank test was conducted to compare the differences in the exhaustion times between the two groups. The significance of the differences between the two groups was determined using Student’s unpaired *t*-test. A *p* value of less than 0.05 was considered significant. Analyses were performed using GraphPad Prism 9.4 software (San Diego, CA, USA) or IBM SPSS 27 software (Chicago, IL, USA).

### 2.6. Additional Methods

Further details of the methods used in the study are provided in the Appendix A.

## 3. Results

### 3.1. HPLC Analysis of JLSE

A typical chromatographic profile of JLSE that includes its main components is provided in Figure 1. The amount of quercetin-3-O-glucuronide (retention time, 4.8 min) was quantitated using its standard as 4.8 mg/g in JLSE.

### 3.2. JLSE Supplementation Enhances Endurance Exercise Performance in Mice

JLSE supplementation over the four week-experimental period did not induce changes in body weight and food intake (Appendix A), nor did it cause liver damage, as serum levels of aspartate aminotransferase and alanine aminotransferase levels were comparable between groups (Appendix A). To evaluate the endurance exercise performance, we measured maximal running time and running distance on the motorized treadmill. The mice in the JLSE-supplemented group showed a significantly enhanced maximum running time, greater running distance, and longer run time to exhaustion than the mice in the control group (Figure 2A–C). Similarly, the grip strength performance in the JLSE group was significantly better than that of control mice (Figure 2D).

### 3.3. JLSE Supplementation Increases Mitochondrial Oxidative Capacity in Mice

To explain the enhanced endurance exercise performance, we performed indirect calorimetric analysis of energy metabolism. Supplementation with JLSE caused a significant decrease in the respiratory exchange ratio (RER and VCO_2_/VO_2_) relative to the control mice (Figure 3A–C and Appendix A), indicating a reliance on fat over carbohydrates as an energy source. Additionally, energy expenditure and heat production tended to be increased by JLSE supplementation (Figure 3D,E and Appendix A).

Consistent with the change in RER, a higher oxygen consumption rate was observed in C2C12 cells treated with JLSE (Appendix A), confirming the effects of JLSE in the shift towards oxidative phosphorylation.

### 3.4. JLSE Supplementation Increases the Proportion of Slow Muscle Fibers in Gastrocnemius Muscles

To investigate whether JLSE enhanced exercise performance by changing the size or proportion of oxidative slow muscle fibers, we evaluated the effect of JLSE on a cross-sectional area of gastrocnemius muscle fibers. We did not observe a difference between the two tested groups (Figure 4A and Appendix A). However, we noted a significant alteration in the composition of muscle fibers between the tested groups: JLSE supplementation increased the proportion of type I and IIa muscle fibers while decreasing the amount of type IIx and IIb (Figure 4B). These changes to fiber type composition were further confirmed by immunostaining of SDH, which is a marker of oxidative slow fibers. Less than 50% of the examined myofibers were SDH positive in the control mice, whereas over 60% of myofibers were SDH positive in the gastrocnemius muscles of the JLSE-supplemented mice (Figure 4C). mRNA levels of slow muscle fiber genes such as *Myh7* and *Myh2* were also significantly higher in the JLSE group than in the control mice (Figure 4D). In summary, these results suggest that JLSE supplementation enhances endurance exercise performance during treadmill exercise training by changing muscle fibers from the fast to slow types.

### 3.5. JLSE Supplementation Enhances Mitochondrial Biogenesis by Upregulating Sirt6

Since oxidative fiber is characterized by its higher mitochondria content, we next determined the effects of JLSE supplementation on mitochondrial count. Our tests showed that JLSE supplementation increased the Ox-Phos protein levels (Figure 5A), the amount of mitochondrial DNA (Figure 5B), and the levels of mRNA involved in mitochondrial biogenesis and oxidative phosphorylation (Figure 5C), all of which suggest an escalation in mitochondrial biogenesis after JLSE supplementation.

Because mitochondrial content is also affected by changes in mitochondrial dynamics [27], we further compared the expression of mitochondrial fusion-fission proteins between groups. JLSE supplementation was observed to have raised mRNA and protein levels of mitochondrial fusion-fission genes (i.e., OPA1, Mfn1, Drp1, and Fis1) to levels higher than those observed in control mice (Figure 5D,E), supporting the hypothesis that JLSE’s effect on mitochondrial content arises as a consequence of both the regulation of mitochondrial biogenesis and dynamics.

We recently identified Sirt6 as playing a critical role in mitochondrial biogenesis in skeletal muscles by increasing the transcription of cyclic AMP response element binding protein (CREB) and by downregulating SRY-box transcription factor 6 (Sox6), a key repressor of slow fiber-specific gene [28]. As CREB is a negative regulator of Sox6 [29], we questioned whether JLSE extract would boost mitochondrial biogenesis by enhancing Sirt6 activity, while concomitantly upregulating CREB and downregulating Sox6 activities. We therefore analyzed the expression of Sirt6, CREB, Sox6, and their downstream signaling proteins that lead to fiber type switching. As shown in Figure 5F, JLSE supplementation increased the Sirt6 and CREB protein levels but decreased Sox6 expression. Additionally, CREB downstream proteins such as PGC-1α and Nor1 were increased by JLSE supplementation (Figure 5F). These results suggest that JLSE supplementation induces slow-twitch oxidative myofiber configuration by regulating the Sirt6-CREB-Sox6 axis.

## 4. Discussion

In this study, we observed that supplementation with JLSE over a four week period augmented endurance capacity and increased proportions of slow and intermediate MyHC isoforms. Exercise-induced endurance parameters, including maximum running time, running distance, and run time to exhaustion, were increased by JLSE supplementation.

Carbohydrates and fat are the two major sources of energy that can be used by well-fed subjects to generate ATP in the mitochondria during aerobic exercise [30]. The relative contribution of each energy source is primarily determined by the intensity and duration of exercise [31]. During high-intensity aerobic exercise (i.e., over 65% of maximal oxygen uptake (VO_2max_)), carbohydrates are the predominant energy source, as they yield more ATP per unit of oxygen than fat. However, during low- to moderate-intensity exercise (~65% VO_2max_), fat is dominant energy source, as it produces a higher amount of ATP when completely metabolized. The specific energy source used is often expressed as an RER value, with an RER value of 1 indicating that a subject predominantly utilizes carbohydrate as a fuel source, while a RER value of 0.7 suggests that fat is the primary fuel source. It is well-documented that aerobic exercise or endurance training increases mitochondrial volume and the capacity for fat oxidation, even despite high carbohydrate intake [32]. Therefore, determining the RER of mice as well as the mitochondrial volume in their myofibers are practical means of evaluating the impact of endurance training or other interventions [33]. In this study, oral supplementation with JLSE resulted in an impressive enhancement to endurance capacity and a significant downregulation of RER in mice. At the tissue and cellular levels, a significant increase in red muscle fibers and simultaneous increase in mitochondrial volume were observed in JLSE-supplemented mice, suggesting that JLSE changed the mice metabolic phenotype to augment exercise performance by altering fuel preference and oxidative machinery. Additionally, muscle plasticity was achieved by the changes to myofiber composition as well as the alterations to the functional property of each myofiber, both of which occurred in response to JLSE supplementation.

Minegishi et al. [26] previously reported that red grape leaf extract increases the expression of enzymes involved in fatty acid oxidation (acetyl CoA oxidase and carnitine palmitoyltransferase 1) and mitochondrial biogenesis (PGC-1α and mitochondrial transcription factor A), observing that the administration of red grape leaf extract over a period of ten weeks increased swimming endurance capacity by 34%. We likewise observed an enhanced running endurance capacity after four weeks of JLSE supplementation. As endurance exercise occurs preferentially in slow oxidative (type 1) and intermediate (type IIa), muscle fibers and is usually accompanied by a fast-to-slow switch in fiber type [2,3], JSLE supplementation may cause changes in muscle fiber type under endurance training. Indeed, a significant increase in the composition of type I and IIa muscle fibers and the number of SDH-positive oxidative fibers was detected in the skeletal muscles of the JSLE-supplemented mice. Slow and fast fibers differ in their capacity to produce ATP. In general, slow fibers have a higher number of mitochondria and denser capillary than fast fibers [28]. Consistent with these well-established facts, we observed that JLSE supplementation increased the number of mitochondria and oxidative capacity, as was reflected in the increased levels of mRNA and proteins associated with Ox-Phos, mitochondrial biogenesis, and mitochondrial dynamics. Together with the observed changes to RER, our results suggest that JLSE supplementation activates slow-twitch contractile machinery by increasing the amount of slow fibers and by enhancing their capacity to burn fat over glucose during long-term endurance training. We also carried out a muscle fiber size analysis that involved measuring cross-sectional areas and found that fiber size in the gastrocnemius muscle was not affected by JLSE supplementation. Accordingly, we conclude that the observed changes in fiber-type composition are not associated with a defect in muscle fiber generation or growth.

We identified quercetin as the primary bioactive substance in the JLSE. Several animal and clinical studies have shown that quercetin treatment increase the mitochondrial biogenesis in skeletal muscles [34,35,36,37]. Of course, an increase in mitochondrial content leads to maximal endurance performance. In contrast to these reports, Koshinaka et al. [38] did not observe positive effects of quercetin on mitochondrial biogenesis in rat skeletal muscle, suggesting that quercetin treatment might provide a disadvantage to muscle adaptation when administered with exercise training. These studies collectively suggest that the beneficial effect of JLSE supplementation on exercise endurance may be due to the combined influence of quercetin along with several other compounds that are contained in JLSE.

The number of mitochondria and their function are determined by a variety of transcriptional regulators encoded by nuclear and mitochondrial DNA [10]. We recently reported that Sirt6, by increasing the transcription of CREB and by downregulating Sox6, is critical to the processes of mitochondrial biogenesis and OxPhos protein induction in skeletal muscles [28]. We therefore analyzed the expression of these proteins in gastrocnemius muscles and found that JLSE supplementation increased Sirt6 and CREB but decreased Sox6 expression. As CREB is a downstream transcriptional regulator of Sirt6 [28] and CREB is a transcriptional repressor of Sox6 [29], JLSE supplementation may have induced slow-twitch oxidative type myofiber configuration through the following cascade: JLSE supplementation → induction of Sirt6 → transcriptional induction of CREB → transcriptional repression of Sox6.

## 5. Conclusions

Reprogramming of fast-to-slow myofiber switch can alleviate the pathogenesis of metabolic diseases, including insulin resistance and fatty liver diseases [7]. Based on the currently reported results, we suggest that a combination of grape by-products (leaf, stalk, or shoot) and endurance training may prevent metabolic diseases by modulating skeletal muscle metabolism. However, we did not include a group of sedentary or anaerobic-exercise-trained mice in this study and can offer only a limited interpretation of JLSE’s effects on endurance exercise. Future clinical studies are also warranted to validate the exercise endurance-enhancing effect of JLSE in humans.

## Figures and Tables

**Figure 1 nutrients-14-04033-f001:**
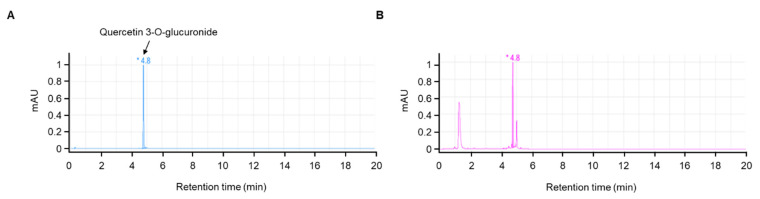
HPLC chromatogram of quercetin-3-O-glucoronide (standard, (**A**)) and ethanol extracts of the JLSE (**B**). A reverse phase column (Phenomenex Kinetex C_18_ 2.6 µm, 2.1 × 50 mm i.d.) was eluted with 0.1% formic acid (flow rate 0.5 mL/min) and monitored at 350 nm. The * peaks were identified by co-injection with quercetin-3-O-glucuronide. mAU, milli-absorbance units.

**Figure 2 nutrients-14-04033-f002:**
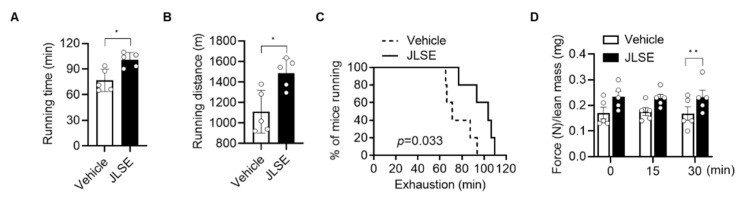
Effects of JLSE supplementation on exercise performance measures. (**A**,**B**) After 4 weeks of treadmill exercise training, running time and running distance were measured (*n* = 5). (**C**) Kaplan–Meier plot showing the running population against time to exhaustion (*n* = 5). Log rank test was used for comparison between two groups (*p* = 0.033). (**D**) Forelimb grip strengths were measured at 15 min intervals and normalized against lean body mass (*n* = 5). Values are mean ± SD. * *p* < 0.05 and ** *p* < 0.01.

**Figure 3 nutrients-14-04033-f003:**
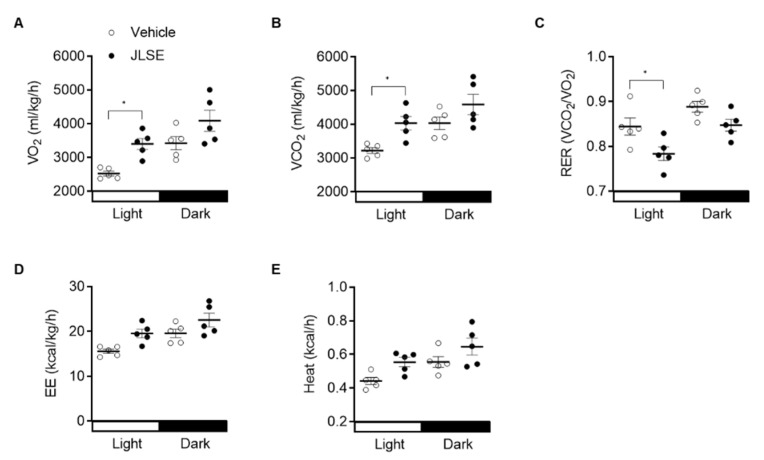
Increased energy expenditure in JLSE-supplemented mice. Indirect calorimetry was performed using an 8-chamber Oxymax system. Mice were acclimatized to cages for 24 h and data were collected for an additional 24 h. (**A**–**C**) Twenty-four hour VO_2_ consumption rates, VCO_2_ production rates, and respiratory exchange ratio (RER) in mice (*n* = 5). (**D**,**E**) Twenty-four hour average energy expenditure (EE) and heat production in mice (*n* = 5). Values are mean ± SD. * *p* < 0.05.

**Figure 4 nutrients-14-04033-f004:**
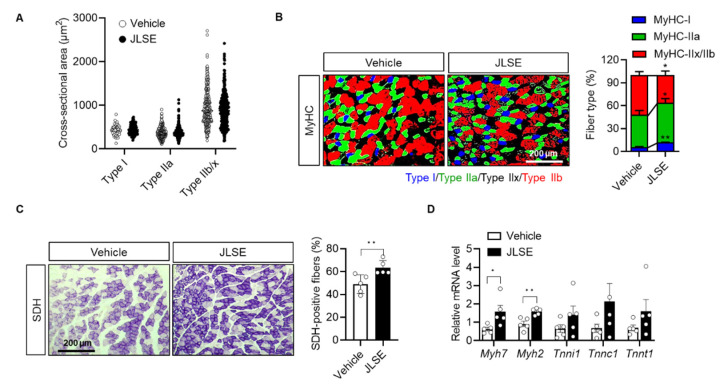
Alteration of myofiber composition in gastrocnemius muscles in JLSE-supplemented mice. (**A**) Cross-sectional area (CSA) of myofibers. (**B**) Immunofluorescence staining for MyHC-I, MyHC-IIa, and MyHC-IIx/IIb. Composition of each myofiber was quantified (*n* = 5). (**C**) Succinate dehydrogenase (SDH) staining and quantification of SDH-positive fibers (*n* = 5). (**D**) Expression of slow fiber genes was compared by qPCR (*n* = 5). Values are mean ± SD. * *p* < 0.05 and ** *p* < 0.01.

**Figure 5 nutrients-14-04033-f005:**
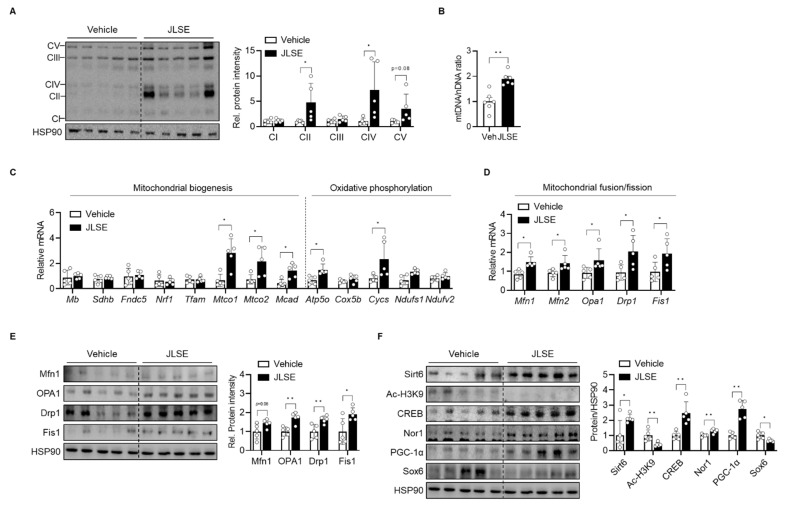
Increase in oxidative capacity in gastrocnemius muscles in JLSE-supplemented mice. (**A**) Western blotting of oxidative phosphorylation complex (*n* = 5). (**B**) Mitochondrial DNA (mtDNA) was quantified by qPCR using nuclear DNA (nDNA) as a standard (*n* = 5). (**C**) qPCR analysis of genes related to mitochondrial biogenesis and oxidative phosphorylation (*n* = 5). Expression of each gene was normalized with housekeeping *Gapdh*, whereas expression of mitochondrial genome-encoded genes *Mtco1* and *Mtco2* was normalized with *16S rRNA*. (**D**,**E**) The mRNA and protein levels of genes involved in mitochondrial dynamics were examined by qPCR and Western blotting, respectively (*n* = 5). (**F**) Western blotting analysis of Sirt6-CREB-Sox6 axis (*n* = 5). Values are mean ± SD. * *p* < 0.05 and ** *p* < 0.01.

## Data Availability

The datasets generated during and/or analyzed during the current study are available from the corresponding authors upon reasonable request.

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
