# Peer review of "Supplementation with Vitis vinifera Jingzaojing Leaf and Shoot Extract Improves Exercise Endurance in Mice"

_nutrients, 2022, doi:10.3390/nu14194033_

Round 1
Reviewer 1 Report
The manuscript "Supplementation with Vitis vinifera Jingzaojing leaf and shoot extract improves exercise endurance in mice" by Lee et al offers some new insights on the mechanism of action by stimulating mitochondrial biogenesis and encouraging switch for fast to slow myofibers. The experiments are well-though and systematic. Although there are some areas of improvement. Here are some comments on how to better the manuscript.
1. Introduction: I suggest that the authors mention about the benefits on increased mitochondrial biogenesis and fast-to-slow myofiber switch on health. What metabolic phenotypes are they associated and how can supplementation of foods that contribute to mitochondrial biogenesis and fast-to-slow myofiber switch be beneficial.
2. Methods: How were the vehicle and JLSE supplemented? Dose and Route? Why was this dose used? The literature mentions at least 4 doses with a variety of measurements that seem to be dose-dependent. In addition, is there any data on measuring oxygen consumption rate of muscle treated with JLSE using seahorse?
3. Statistics: There are two groups (vehicle and JLSE) so I am wondering when one-way ANOVA was used and what was the reason? Figure 2C must be analyzed using a Kaplan-Myer survival analysis.
In addition, the sample size is low (n=5/group). Can you provide a power analysis calculation.
3. How many stained muscle section per animal was analyzed for Fig 4B-C? Why were all samples not included for the WB in Fig 5A, E and F?
4. A few insights for the discussion: Based on Fig.1, much of the JLSE is quercetin. Is the effect of JLSE practically the effect of quercetin. Studies have shown the quercetin has can stimulate mitochondrial function. Also, it would be ideal to discuss the effect of JLSE on mitochondrial function and muscle physiology on a resting or anaerobic exercise. Is taking quercetin ideal that JLSE in promoting mitochondrial biogenesis and fast-to-slow myofiber switch?
Author Response
1. Introduction: I suggest that the authors mention about the benefits on increased mitochondrial biogenesis and fast-to-slow myofiber switch on health. What metabolic phenotypes are they associated and how can supplementation of foods that contribute to mitochondrial biogenesis and fast-to-slow myofiber switch be beneficial.
Response: We described the physiologic and clinical importance of myofiber specialization and plasticity and the way to induce fast-to-slow myofiber switch by food supplementation in the Introduction section:
2. Methods: How were the vehicle and JLSE supplemented? Dose and Route? Why was this dose used? The literature mentions at least 4 doses with a variety of measurements that seem to be dose-dependent. In addition, is there any data on measuring oxygen consumption rate of muscle treated with JLSE using seahorse?
Response: Dose and route of JLSE supplementation are shown in Figure S1. In this study, we chose the doses of JLSE at 300 mg/kg because, in previous reports (references #19, 20, and 22), oral treatment of mice or rats with grape leaf extract at doses of 200 and 500 mg/kg had beneficial effects in various disease models. Seahorse analysis data were added in Figure S4.
3. Statistics: There are two groups (vehicle and JLSE) so I am wondering when one-way ANOVA was used and what was the reason? Figure 2C must be analyzed using a Kaplan-Myer survival analysis. In addition, the sample size is low (n=5/group). Can you provide a power analysis calculation.
Response: We apologize for our negligence in this regard. We analyzed the data using Student’s unpaired t-test. As commented, Kaplan-Meier survival curve was used to estimate the percentage of mice running until exhaustion. Log-rank test p value was 0.033. The results of power analysis calculation by using Logrank Power package in R software (verson 4.0.3) for log-rank test was 0.08 (Total sample size=10, type 1 error=0.05, and effect size=0.3).
4. How many stained muscle section per animal was analyzed for Fig 4B-C? Why were all samples not included for the WB in Fig 5A, E and F?
Response: The sample size for Figs. 4B-4C is n=5/group, which is presented in Figure 4 legend. As commented, we included all samples for Western blotting analysis in Figs. 5A, 5E, and 5F.
5. A few insights for the discussion: Based on Fig. 1, much of the JLSE is quercetin. Is the effect of JLSE practically the effect of quercetin. Studies have shown the quercetin has can stimulate mitochondrial function. Also, it would be ideal to discuss the effect of JLSE on mitochondrial function and muscle physiology on a resting or anaerobic exercise. Is taking quercetin ideal that JLSE in promoting mitochondrial biogenesis and fast-to-slow myofiber switch?
Response: As commented, we discussed the conflicting effects of quercetin on mitochondrial biogenesis and exercise endurance in the Discussion section. Because we did not include a group of sedentary or anaerobic exercise mice in this study, we can offer only a limited interpretation of the combining effect of JLSE and endurance exercise compared to exercise-only effect. We described this point in the Conclusion section as a limit of this study.
Reviewer 2 Report
This is a very interesting article about supplements for exercise endurance improvement in mice.
It is very well written and I particuarly appreciate figures and tables
Minor suggestions
- pay attention to english grammar and syntax rules all over the manuscript
- please consider the following reference when speaking about supplementations and microscopic cellular changes (http://www.mltj.online/influence-of-supplements-and-drugs-used-for-the-treatment-of-musculoskeletal-disorders-on-adult-human-tendon-derived-stem-cells/) in the introduction section
- please shorten the conclusion section, since only main results of your study, along with some future suggestions, should be included
Author Response
1. Pay attention to English grammar and syntax rules all over the manuscript
Response: This manuscript is edited by native English speakers. We have attached the certificate for English editing.
2. Pease consider the following reference when speaking about supplementations and microscopic cellular changes (http://www.mltj.online/influence-of-supplements-and-drugs-used-for-the-treatment-of-musculoskeletal-disorders-on-adult-human-tendon-derived-stem-cells/) in the introduction section
Response: As commented, we have cited suggested reference in the Introduction section.
3. Please shorten the conclusion section, since only main results of your study, along with some future suggestions, should be included
Response: As commented, we rephrased the conclusion section.
Reviewer 3 Report
Manuscript entitled “Supplementation with Vitis vinifera Jingzaojing leaf and shoot extract improves exercise endurance in mice” by Lee et al observed the mice supplemented with Vitis vinifera Jingzaojing leaf and shoot extract and conducted an experiment, such as exercise endurance, mitochondrial oxidative capacity. In addition, Authors also demonstrated the fast-to-slow myofiber switching by measuring the expression of sirtuin 6 and Sox6. Though outcome of the study were clearly represented with figures and the results were discussed well, I have few minor comments below.
Include notation of the primer sequences in supplementary materials.
Figure S3- Though I can understand grey and white box represents dark-light cycle, mention them in caption. Figure panels should be corrected to A, B, C and D instead A, B, D and E in Figure S3 as per the caption.
Figure S1 depicts the animal adaption, treadmill acclimatization and JLSE administration. However, It was not clear when the RER and histopathology was conducted? Why does RER was not done before the start of the experiment?
For me, it looks like the plot for the figures, S3B, S3D and S3E (as per the notation of the panel) are similar, only y axis is changed. Please check with the data.
Discussion: discussion of the outcome might have included with few more reference citations.
Line 286-291: Makes confusion with the current study. Hence, Move them in to either introduction or discussion section.
Author Response
1. Include notation of the primer sequences in supplementary materials.
Response: As commented, we notated the primer sequences in Table S1.
2. Figure S3- Though I can understand grey and white box represents dark-light cycle, mention them in caption. Figure panels should be corrected to A, B, C and D instead A, B, D and E in Figure S3 as per the caption.
Response: Thank you for pointing out our mistake.
3. Figure S1 depicts the animal adaption, treadmill acclimatization and JLSE administration. However, It was not clear when the RER and histopathology was conducted? Why does RER was not done before the start of the experiment? For me, it looks like the plot for the figures, S3B, S3D and S3E (as per the notation of the panel) are similar, only y axis is changed. Please check with the data.
Response: We described the time point of indirect calorimetry analysis and tissue collection for histological analysis in the Figure S1 legend. We carefully looked at the raw data of Figs. S3A-S3D. It looks pretty similar pattern, but different data.
4. Discussion: discussion of the outcome might have included with few more reference citations.
Response: As commented, we have cited appropriate references.
5. Line 286-291: Makes confusion with the current study. Hence, Move them in to either introduction or discussion section.
Response: Reviewer 2 commented similar point. We rephrased the conclusion section.